# Ethnic inequality and forced displacement

**Moritz Hennicke**[1,2]*, **Tilman Brück**[3,4,5]

**1** Université libre de Bruxelles, Brussels, Belgium, **2** Paris Cergy Université, Cergy, France, **3** ISDC - International Security and Development Center, Berlin, Germany, **4** Natural Resources Institute, University of Greenwich, London, United Kingdom, **5** Leibniz Institute of Vegetable and Ornamental Crops, Großbeeren, Germany

* moritz.hennicke@ulb.be

**Data Availability Statement:** Access to the 2010 wave of the LiK study is subject to a contract between the user and IZA. The contract can be requested through following links: https://datasets.iza.org/dataset/download/124/life-in-kyrgyzstan-study-2010-2019 http://dx.doi.org/10.15185/izadp.7055.1 Education by ethnicity in 2009 at the

## Abstract

How does ethnic inequality shape victimization in violent conflicts? Our case study of the 2010 conflict in Kyrgyzstan tests whether communities with higher ethnic inequalities in education experienced more intense displacement. We find that local inequality in education between Kyrgyzstan's ethnic majority and its largest minority robustly predicts patterns of forced displacement, controlling for alternative approaches like ethnic spatial segregation or polarization. By decomposing inequality, we observe that local educational advantage towards the other ethnic group is associated with a lower likelihood of displacement. Ethnic Uzbeks with low education levels relative to Kyrgyz in their area have the highest displacement rate.

## 1 Introduction

Kyrgyzstan has been throughout its history a place many ethnicities called their home. Kyrgyzstan ranks 12th out of 159 countries in James Fearon's global cultural diversity index [1]. In 2010 the young country was rocked by violent ethnic conflict. A short period of violence left more than 400 dead and temporarily displaced some 200,000 individuals, mostly ethnic Uzbeks [2]. Riots erupted along ethnic lines between the ethnic Kyrgyz majority and the ethnic Uzbek minority. International media reported about slogans such as "Death to the Uzbeks" spray-painted on wrecked buildings [3].

A long-standing paradigm in the social sciences has proposed that inequality can spur violent conflict. One group of scholars stresses the importance of between-group inequalities, often described as *horizontal inequalities*, for the *onset* of conflict [4, 5]. Recent cross-country evidence has made a strong case for an emphasis on inequality within ethnic groups, *vertical inequality*, for the *intensity* of conflict [6]. Against this background, it remains unclear who is affected by the conflict when inequality drives it. How does inequality among people of different ethnic groups shape forced displacement?

We address this question by studying the relationship of ethnic inequality on forced displacement. We draw on data from the nationally representative Life in Kyrgyzstan (LiK) Study in 2010, the 2009 Census and the Uppsala Conflict Data Program [7–9]. Relying on nationally representative survey data for the displacement outcome offers the advantage that there are no biases common in administrative data in conflicts [10].

community level can be obtained by request from the National Statistics Committee and their population and demographics team. Contacts can be found on this website: http://www.stat.kg/en/statistics/naselenie/ UCDP Georeferenced Event Dataset (GED) Global version 21.1 can downloaded through the UCDP website: https://ucdp.uu.se/downloads/index.html#ged_global The authors confirm that others would be able to access these data in the same manner as the authors. The authors also confirm they did not have any special access privileges that others would not have.

**Funding:** The authors received no specific funding for this work.

**Competing interests:** The authors have declared that no competing interests exist.

Using our data, we document that displacement rates of Uzbeks are higher than of Kyrgyz and spatially concentrated close to the border with Uzbekistan. Variation across communities demonstrates how places with higher disparities in education between Kyrgyz and Uzbeks experienced higher displacement rates. In particular the role of *horizontal inequality* emerges as relatively robust, compared to alternative drivers of conflict proposed in the literature such as ethnic polarization and spatial segregation [11, 12].

We complement the analysis of variation at the group level with individual regressions. We identify the victims and their position in the local social hierarchy of their community. We estimate linear probability models, regressing forced displacement on the individual rank in the regional distribution of ethnicity and education. By doing so, we decompose horizontal inequality to investigate whether advantage or disadvantage towards other people of different ethnicities can explain displacement. We find that educational advantage towards the other ethnic group in the area was associated with a lower probability to flee. The associations suggest that not education per se explains displacement, but rather the fact that an individual possessed more or less education than most people of the other ethnicity in the area. We find that horizontal inequality is also directly associated to displacement, when controlling for the proximity to casualties. It is likely that both horizontally advantaged individuals had a lower subjective and objective personal risk of getting harmed during the conflict.

We make three contributions to the study of inequality and conflict with this descriptive case study. Firstly, our paper is the first to address this topic in the context of Central Asia. Secondly, we jointly test inequality indices against polarization and geographic segregation. Thirdly, we link the group and individual level by investigating the correlation of individual rankings in local group distributions with individual conflict outcomes.

Several limitations remain to this paper. We rely on survey data collected after the conflict. This implies, firstly, that we can only provide anecdotal evidence by the UNHCR that displaced populations across the borders had returned before the field work of the survey. Secondly, the place of residence after the conflict might not coincide with the place of residence during data collection for internally displaced. We offer a remedy against this measurement problem by testing the robustness of our findings with a subset of individuals who reside in the place where they were born. And finally, we cannot exploit exogenous variation in ethnic inequalities such that confounders remain problematic. We report a battery of sensitivity checks and find that unobservables as strong as the distance to the Uzbek border are not sufficient to explain away the observed estimate. We therefore interpret our estimates as meaningful correlations given the overall robustness to alternative measurements and samples.

The following Section sketches the background of the conflict. Section III introduces our theoretical framework guiding our empirical analysis. Section IV presents our data and methodology, section V summarizes our results and Section VI concludes.

## 2 Background of 2010 conflict

The Ferghana Valley, situated around the border triangle of the post-Soviet states of Kyrgyzstan, Uzbekistan and Tajikistan, has historically been a place of cultural and linguistic diversity. In the Kyrgyz part of the valley, ethnic Uzbeks nowadays constitute the second largest ethnic group.

Historians describe the Soviet Union's system of political rule as ethnofederalism. An ethnic group was supposed to have one single homeland, the so called *rodina* ruled by local co-ethnic elites [13]. As a result, large ethnic Uzbek groups living outside of Uzbekistan in contemporary Kyrgyzstan were politically underrepresented and did not possess any form of political autonomy. With the demise of the Soviet Union and its institutional coercion and control,

Kyrgyzstan as one of the ethnically most heterogeneous former Soviet states was left to itself to form its own sovereign state. The country's politics embraced a search of cultural identity within its national borders. The Uzbek communities in Southern Kyrgyzstan have since then been struggling to define their position between a newly drawn national border to their South and young transitioning institutions in Kyrgyzstan's northern capital Bishkek [13].

Prior to the disintegration of the Soviet Union, there were two major violent conflicts between Kyrgyz and Uzbeks in the southern regions of Kyrgyzstan. The first episode took place in June 1990, triggered by struggles over political power, increased social disparities, high unemployment rates, and disputes over land distribution, as scholars of the conflict claimed [14, 15]. This violent episode took 170 lives and more than 5,000 were injured in shootings, fights and burnings. Uzbeks protested over their under-representation in the public and political sphere while Kyrgyz were dissatisfied with their economic situation and land shortages. The violence was finally halted by army troops that were employed in the conflict region. The perpetrators of violent attacks were prosecuted. Since the 1990 episode of violence, communities with a high percentage of Uzbeks have been potential conflict areas, and disputes over land and water distribution are increasingly reported to have an ethnic dimension [13, 16–18].

The second major inter-ethnic clashes between Uzbeks and Kyrgyz took place in spring 2010. Observers of the conflict describe the context as follows: the global financial crisis in 2009 created pressure on the Kyrgyz economy, with remittances from Russia declining [18]. People in Kyrgyzstan were increasingly dissatisfied with high corruption, increased energy tariffs, media surveillance, and nepotism [19]. At the same time, President Bakiyev's government increasingly persecuted influential opposition leaders and journalists [15]. In April 2010, violence erupted in the city of Talas and in the capital city of Bishkek. The center of the tension then shifted towards southern Kyrgyzstan after Bakiyev left Bishkek for his hometown Jalalabad, and an interim government was established. While Uzbeks asked for more political representation in the government and the recognition of the Uzbek language, the new draft constitution from May 2010 did not address these points [15]. Kyrgyz gangs clashed with ethnic Uzbeks gangs between June 10 and June 14, 2010. Information and rumors about the inter-ethnic clashes spread rapidly to other provinces and cities, and the number of violent attacks further increased [15]. According to anecdotal evidence, property of Uzbek households was disproportionately attacked and more Uzbeks relative to Kyrgyz were arrested [15, 18].

UN sources conclude that more than 400 people were killed (the majority of them Uzbeks), 2500 people were injured, approximately 75,000 fled to neighboring Uzbekistan and 300,000 were internally displaced [2, 20]. The Kyrgyz authorities did not release an ethnic breakdown of deaths. Approximately 2,800 mostly private buildings were damaged or totally destroyed [21]. Even though the period of violence was short-lived, recent evidence suggests that the conflict had a long lasting detrimental impact on prosociality in affected areas in Osh [22]. Against this background we test how ethnic inequality relates to victimization.

## 3 Theoretical background

This paper builds on a rich theoretical and empirical literature on ethnic and socioeconomic inequalities, violent conflict and forced displacement. To guide our empirical analysis of inequalities and displacement, we encode our assumptions of the underlying causal process in a directed acyclic graph (DAG) based on our reading of the existing literature. Fig 1 contains this graph, where nodes are random variables and directed edges trace the causal relationship going from one variable to another. Inequalities, violence and displacement are observed as are the covariates X. The main variable of interest are connected through mechanisms M1, M2

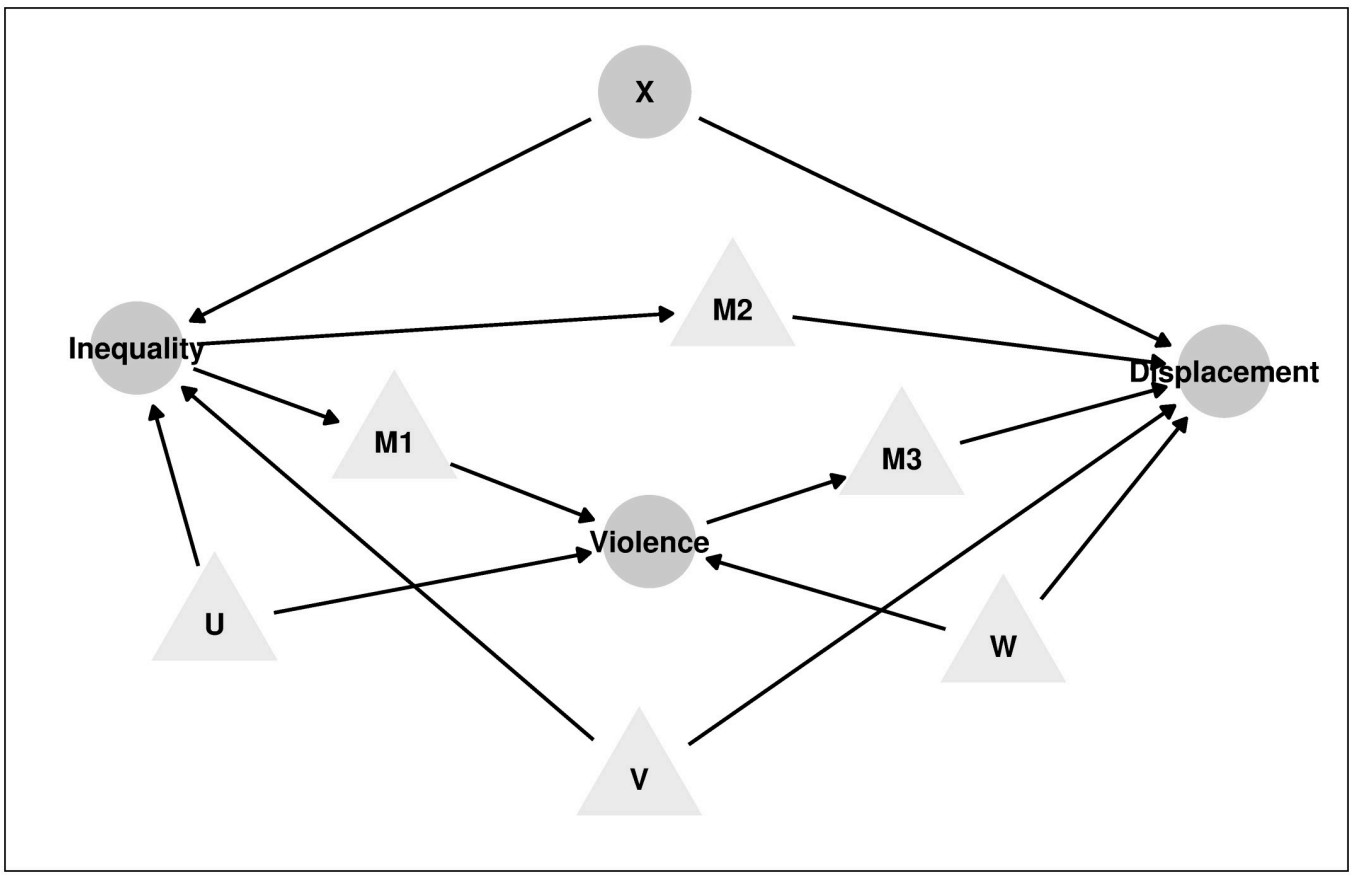

**Fig 1. Directed acyclic graph of causal relationships. Round Nodes**: Observed variables. **Triangular Nodes**: Unobserved variables. **Edges**: Direction of causal relationship.

and M3. Unobserved confounders are denoted with U, V and W, which embody the threats to identification.

At the root of this causal graph is the relationship between ethnic and socioeconomic divisions and violent conflict (ethnic inequalities → violence) which operates through some mechanism M1. The literature makes the distinction between within-group and between-group inequalities. Recent work finds a positive association of within-group inequalities with conflict onset and intensity respectively [6, 23]. Some scholars find support for a positive association of between-group inequalities with conflict in cross-country variation [4, 5], and negative association in India [24]. Socioeconomic inequalities are measured in the literature along the dimension of income (e.g. [6]), economic activity with night lights (e.g. [5]), wealth, and education (e.g. [4]).

We measure inequalities in the dimension of education for the pragmatic reason that we do not have pre-conflict information on income or wealth. One concern about this graph might be the directionality of the relationship between conflict and education [25]. As we are not aware of any research demonstrating a response of education to the expectation of future conflict, we think that completed education pre-conflict might not suffer from simultaneity bias. We think that horizontal inequality in education might be linked to violence through several unobserved mechanisms (M1): Firstly, the collective experience of discrimination in terms of access to education might be conducive to collective organization against the other privileged

ethnic group [26]. Grievances could even be higher when imbalances arise in a publicly funded education system. Secondly, when some form of social class identity acquired with schooling overlaps with ethnic identity, more profound identitarian markers for division exist. Thirdly, opportunity costs of engaging into violent acts such as looting might be lower for horizontally disadvantaged individuals.

A strand of the literature has looked at the effects of violent conflict on displacement (`violence → displacement`) via mechanism M3 [27]. On the one hand, civilians might flee out of fear or as a consequence of minimizing risks of harm. Civilians might also follow more forward-looking motives with resettling strategically to escape future expected targeting by armed groups [28]. On the other hand, civilians might decide to stay and fight, or work in lucrative sectors of the war economy. They might decide to stay in order to fight for some cause, join some armed group for personal protection [29], or for personal gain of wealth, assets and power [30].

We conjecture that, besides observed violence, a direct connection (`ethnic inequalities → displacement`) exists and operates through mechanism M2. We propose two alternatives to guide our empirical work. Firstly, the decision to flee might actually be taken based on a subjective perception of risk. Horizontal inequality measures a reality of social hierarchy between individuals of different ethnicities, which might shape the individual belief of the risk to fall victim to aggressions by members of the other ethnic group regardless of observable violence. Secondly, many forms of unobserved violence might exist such as sexual abuse, psychological aggression and socioeconomic discrimination which might not correlate perfectly with observed violence such as fighting. Survey data might not be able to measure these acts as victims might find it difficult to speak precisely about these issues due to stigmatization.

Other observed characteristics $X$ determine ethnic inequalities and displacement. The distance to the Uzbek border, decreased the cost of flight for ethnic Uzbeks across the border and shaped socioeconomic opportunities for ethnic Uzbek and Kyrgyz differently in the past. Moreover, we control for varying sets of covariates shown to be important in conflict settings such as population, birthplace, unemployment [31], urban [32], major manufacturing plants [25] and landownership [33].

Unobserved factors affect $U$, $V$ and $W$ our main variables of interest. For instance, we cannot assess whether urban/rural can control for variation in public space across communities, as is argued that they were prone to conflict action in Osh City [34]. Moreover, we cannot control for capacities of local political leaders and elites to engange with rioters and broker deals with gangs [35].

As we cannot exploit an experimental shift in ethnic inequalities, we cannot exclude that these unobserved factors might bias our coefficients. What we are able to do is to test the sensitivity of our results in the presence of unobserved confounders, bounding the potential strength of confounders with major covariates. It should be emphasized that the strong assumption of no confounding is likely to be violated, such that we interpret the estimated coefficients as meaningful correlations.

Moreover, due to limitations of what we can observe, we cannot test for all potential mechanisms reflected upon theoretically. In our community regressions mechanisms are implicit in the total correlation of ethnic inequalities with displacement. At the individual level, we are able to decompose inequality to understand associations with inequality better from an individual perspective.

Finally, in our main specifications we do not control for violence in the regression of displacement on ethnic inequalities. When interested in the total association, the back-door criterion necessitates the exclusion of descendants of the treatment on the path to the outcome

[36]. We include violence in alternative specifications at the expense of inviting confounders `W` along the path (`ethnic inequalities → violence ← W → displacement`).

## 4 Data and methodology

### 4.1 Displacement data

Our main source of data is the 2010 wave of the Life in Kyrgyzstan (LiK) study collected 3 to 6 months after the violent conflict in June [7]. It provides data for households and individuals since all adult members of each household were interviewed separately. Kyrgyzstan is organized administratively into 9 *provinces* (oblasts) including the cities of Osh and Bishkek. Provinces are further divided into 51 *districts* (rayons). We make use of the fact that LiK communities were sampled based on the 2009 Census to merge it back with Census data [8] at the level of 120 *communities*. Our main outcome of interest is the displacement experience of households and individuals during the conflict. The displacement data offers the advantage that we do not rely on administrative sources often politicized or with incomplete coverage. For instance, surveys conducted in Columbia revealed that government programs do not reach the entire population at need [10].

The UNHCR estimated the population of temporarily displaced during the conflict at around 200,000 or 3.7% of Kyrgyzstan's total population at the time [2]. The estimate is likely to be imprecise. Using the nationally representative LiK study with a total number of 7,897 individuals, no over-sampling of the temporarily displaced population took place. The enumerators eventually surveyed a total of 202 individuals who indicated to have experienced displacement during the conflict. This amounts to approximately 2.94% of the whole population or to a total of 157,562 individuals (with the 2009 Census as base population). The survey indicates that 41.15% of the temporarily displaced were Kyrgyz (1.59% of Kyrgyz population) and 54.1% were Uzbeks (11.38% of the Uzbek population). We complement our data with geocoded information on violent incidents from the Uppsala Conflict Data Program such that we can compute distances between communities and violence [9].

Fig 2 shows a map of Kyrgyzstan, where circles illustrate the location of sample communities and the spatial distribution of displacement. Displacement is concentrated in the South

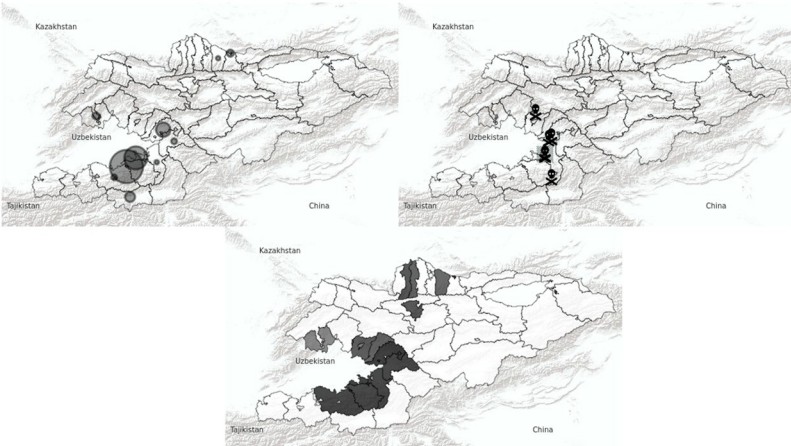

**Fig 2. Maps of local forced displacement, deaths and horizontal inequality in 2010.** Upper left pane: Forced displacement (Source: [7]). Upper right pane: Estimated casualties (Source: [9]). Lower pane: Horizontal inequalities (Source: Own calculations based on [37]).

**Table 1. Forms of displacement.**

| Type of displacement | Ethnic Kyrgyz | | Ethnic Uzbek | | Total | |
|---|---|---|---|---|---|---|
| | No. | % | No. | % | No. | % |
| In same town/village w. friends/family (IDP) | 22 | 11.5% | 49 | 25.5% | 71 | 37.0% |
| Elsewhere in Kyrgyzstan w. friends/family (IDP) | 61 | 31.8% | 6 | 3.1% | 67 | 34.9% |
| Displacement camp within Kyrgyzstan (IDP) | 0 | 0.0% | 29 | 15.1% | 29 | 15.1% |
| Uzbekistan (Refugee) | 0 | 0.0% | 21 | 10.9% | 21 | 10.9% |
| Elsewhere | 1 | 0.5% | 3 | 1.6% | 4 | 2.1% |
| Total | 84 | 43.8% | 108 | 56.2% | 192 | 100.0% |

*Source*: Life in Kyrgyzstan (LiK) study 2010 [7].

around the urban agglomerations of Osh and Jalal-Abad, but not exclusively, as the instances of reported displacement in the North of the country show.

The survey elicits displacement status during the episode of fighting with the question: "In June 2010, did you have to leave your place of living?". We can furthermore make a distinction of the form of displacement with the follow-up question: "Where did you stay most of the time during the absence?". As Table 1 illustrates, Kyrgyz displacement was predominantly internal with 73% of displaced residing elsewhere in Kyrgyzstan and 26% in the same village but with relatives and friends. 19.4% of displaced Uzbeks left for Uzbekistan, 27% stayed in IDP camps within Kyrgyzstan and 45% stayed with friends and relatives in their village. Displacement is significantly higher for Uzbeks than for Kyrgyz as summary statistics in S2 Table in the S1 Appendix show. According to the UNHCR the estimated 75,000 refugees who fled across the border to Uzbekistan had returned to Kyrgyzstan by the 27th June [38]. Nevertheless, an obvious limitation is that our survey only encompasses returnees to Kyrgyz territory. If they had returned in later periods we are unfortunately not able to identify them as part of the displaced population. Another concern with LiK is that individuals did not return to their homes but were surveyed elsewhere in Kyrgyzstan. We can address this concern by sub-setting our sample to individuals, who indicate that they have always lived in the community they were surveyed in.

## 4.2 Measures of local ethnic-inequality

As a first step, we explore the link between displacement, ethnicity and inequality with indices that we construct at the community-level. We focus on education to measure socioeconomic disparities. We construct a pre-conflict index with 2009 Census data measuring the inequality between ethnic Kyrgyz and Uzbeks in terms of secondary school completion at the community level. We follow the literature by defining our measure of horizontal inequality in Eq 1, with $s_{ky,j}$ as the share of secondary educated ethnic Kyrgyz and $s_{uz,j}$ of ethnic Uzbeks in community $j$ [4].

$$HI_j = 1 - exp(-|ln(s_{ky,j}/s_{uz,j})|). \tag{1}$$

The index varies between 0 for communities with equal secondary education shares of both groups and 1 for distributions where everyone is secondary educated in one group and no one in the other. The measure is indifferent to which group has the higher share. Table 2 shows at the province-level, that ethnic Kyrgyz have higher a share of secondary education in 3 provinces, ethnic Uzbeks in 5 provinces and in the province of the capital city Bishkek shares are about the same. The horizontal inequality index ranges from 0 in Bishkek to 0.23 in Osh City.

**Table 2. Summary statistics across provinces.**

|  | Issyk-Kul | Djalal-Abad | Naryn | Batken | Osh | Talas | Chui | Bishkek | Osh City |
|---|---|---|---|---|---|---|---|---|---|
| Ethnic Polarization (index, 0–1) | 0.467 | 0.451 | 0.065 | 0.385 | 0.237 | 0.278 | 0.834 | 0.855 | 0.968 |
| Kyrgyz secondary education (%) | 0.609 | 0.684 | 0.648 | 0.683 | 0.692 | 0.663 | 0.637 | 0.760 | 0.747 |
| Uzbeks secondary education (%) | 0.668 | 0.659 | 0.770 | 0.688 | 0.675 | 0.722 | 0.697 | 0.760 | 0.576 |
| Multi-group HI education (index, 0–1) | 0.022 | 0.033 | 0.006 | 0.011 | 0.027 | 0.095 | 0.064 | 0.013 | 0.124 |
| Uzb-Kyr HI education (index, 0–1) | 0.013 | 0.075 | 0.000 | 0.059 | 0.087 | 0.000 | 0.023 | 0.000 | 0.229 |
| Gini education across groups (index, 0–1) | 0.104 | 0.113 | 0.078 | 0.079 | 0.103 | 0.091 | 0.124 | 0.112 | 0.115 |
| Gini education within Uzbeks (index, 0–1) | 0.100 | 0.092 | 0.090 | 0.091 | 0.089 | 0.080 | 0.093 | 0.073 | 0.087 |
| Gini education within Kyrgyz (index, 0–1) | 0.101 | 0.113 | 0.076 | 0.083 | 0.115 | 0.090 | 0.105 | 0.108 | 0.095 |
| Share forced displaced | 0.000 | 0.033 | 0.000 | 0.040 | 0.030 | 0.000 | 0.003 | 0.001 | 0.257 |

*Note*: Kyrgyzstan is organized administratively into 9 provinces and 51 districts.

*Source*: Life in Kyrgyzstan (LiK) study 2010 [7] and the 2009 Census of the Statistical Committee of the Kyrgyz Republic [37].

In conjunction, we include measures of vertical inequality. We rely on Gini indices, such that we use individual education levels of adults from the LiK study. We use our survey data to compute Gini indices of years of completed education for each community. We repeat this for only Kyrgyz and Uzbeks individuals and obtain three different measures at the community level: Educational inequality across the whole community population, within the Krygyz majority and within the Uzbek minority. We impute the country-wide Gini for communities without an Uzbek or Kyrgyz population, respectively.

The ethnic composition alone can spur conflict, in particular where two ethnic groups are of similar size [11, 39]. We borrow a measure of ethnic polarization from the literature [40], where $n_{r,j}$ represents the share of ethnicity $e$ in community $j$ in $RQ_j = 4 \sum_e^E (1 - n_{e,j}) * n_{e,j}^2$. We use ethnic Kyrgyz, Uzbeks, Russians and Others (Dungans, Uigurs, Tajiks, and Kazakhs). It ranges for provinces from 0.07 (almost no diversity), over medium values of 0.446 in Djalal-Ab (one majority group, some minorities) to 0.968 in Osh City province (almost equal shares for Uzbeks and Kyrgyz).

Our work relates to past work that systematically measure differentials in welfare between both ethnic groups [41]. They find that Kyrgyz have a slightly higher level of expenditure in urban areas combined with higher educational achievements. Table 2 shows the largest differences in Uzbek and Kyrgyz education in the HI index in the Southern regions Djalal-Abad, Batken, Osh and Osh City. Fig 2 shows at a more granular level for communities that province-averages still underestimate the local disparities in education that exist between both ethnic groups.

Beyond regional variation of group indices, we exploit individual responses from the LiK study. We therefore transform individual education levels into completed years of schooling. Fig 3 shows the estimated proportions of ethnicity and education for those who experienced displacement and those who did not. Firstly, ethnic Russians and other ethnicities not reported here for this reason, were almost not affected by displacement. Secondly, the percentage of Uzbeks compared to Kyrgyz among the displaced is higher. Thirdly, the Kyrgyz education

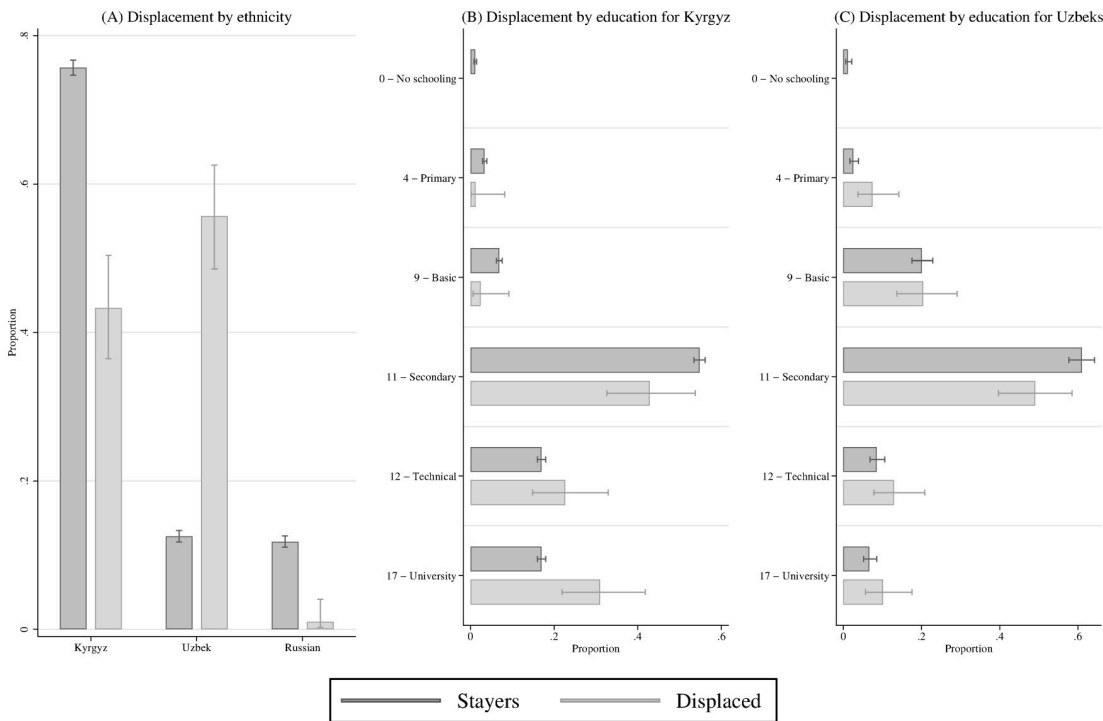

**Fig 3. Education by ethnicity and displacement.** Notes: Cross-tabulations of education, ethnicity and displacement status for the three largest ethnic groups. The left pane shows estimated proportions of ethnicities for the displaced and non-displaced. The middle pane shows estimated proportions of education levels for displaced and non-displaced only for ethnic Kyrgyz. The right pane shows estimated proportions of education levels for displaced and non-displaced only for ethnic Kyrgyz. Error bars indicate 95% confidence intervals. (Source: Life in Kyrgyzstan (LiK) study 2010 [7]).

distribution for the displaced leans further to the right with proportionally more educated members leaving.

## 5 Results

### 5.1 Community cross-section

As a first step, we regress in Eq 3 the share of the population forcefully displaced during the conflict in community $j$ on ethnic inequality measures and controls. As the variance of the displaced proportion decreases closer to 0, we use robust standard errors suitable for smaller sample sizes throughout [42].

$$\text{Forced Displacement}_j = \alpha + \delta \text{ Ethnic Inequality}_j + \beta X_j + \epsilon_j \tag{2}$$

The main explanatory variables of interest are *horizontal* and *vertical inequality* and polarization indices. We include a battery of (in the short-term) time-invariant community characteristics: the fraction of individuals born in the community, a dummy for rural/urban and for major industries from the LiK survey. Additionally, community population and employment are taken from the 2009 Census. The map in Fig 2 shows how proximity to the border with Uzbekistan is associated with displacement. The distance to the Uzbek border should increase the costs of movements of ethnic Uzbeks between Uzbekistan and Kyrgyzstan. Refugee camps, relatives and language at the other side of the border very likely facilitated the movement for

ethnic Uzbeks. For most of the time the border was open and refugee camps were constructed for fleeing ethnic Uzbeks. Moreover, the distance to the Uzbek border likely created differential socioeconomic opportunities for Uzbeks and Kyrgyz historically. We therefore control for the distance to border throughout all specifications.

Our identification strategy for the coefficients of our regression is based on simple between-variation across communities. In the face of multiple threats to identification, we emphasize that our regression coefficients should be interpreted as meaningful correlations.

Table 3 reports coefficients from linear regressions. Controlling for fundamental community characteristics, column A exhibits a sizable and significant correlation between the fraction of displaced and *horizontal inequalities* (HIs). The association is sizable in magnitude as an increase in the index by one standard deviation is associated with an average increase of $0.084^{*}0.54 = 4.5pp$ in the displaced local population. The correlation between forced displacement and horizontal inequality is plotted in S1 Fig in S1 Appendix. Horizontal Inequality and Forced Displacement in the S1 Appendix.

Column B includes the Gini coefficient of education for each community. Interestingly, horizontal inequality does not seem to spatially coincide with *vertical inequality* as the HI coefficient maintains its significance and magnitude. We include Gini indices for the subpopulation of each ethnicity instead of the Gini index across all groups in column C. Neither Kyrgyz nor Uzbek *within-group vertical inequality* is significantly correlated with displacement.

Is the ethnic composition alone of communities associated with forced displacement? Column D tests this question by adding the polarization index, indicating how comparable in size the population of ethnic Kyrgyz and Uzbeks are. Controlling for polarization increases the coefficient of HI by 7pp and it remains significant.

The last column of Table 3 extends the *horizontal inequality* index to multiple groups including Russians, Dungans, Uigurs, Tajiks, Kazakhs and Other. We borrow an index from the literature [43], defined for every community $j$ as $mHI_j = \frac{1}{\bar{y}_j} \left( \sum_e^E n_{e,j} (y_{e,j} - \bar{y}_j)^2 \right)^{1/2}$, where $n_{e,j}$ is the fraction of ethnicity $e$ of the total population and $y_{e,j}$ the mean education level. It increases whenever an ethnicity's group size is large and its average educational attainment drifts away from the community average. The coefficient of the multi group HI is not significant, which gives support to the narrative that clashes were predominantly between ethnic Kyrgyz and Uzbeks groups.

We estimate the direct and indirect association through distance to violence of forced displacement with inequalities in Table 4. The correlation between HIs and displacement decreases by 6pp, which could be interpreted under the strong assumption that no other confounders correlated with violence and displacement introduce bias, that most of the total association between HIs and displacement is indeed direct.

Columns A and B of Table 5 split up the outcome into the share of displaced for the Kyrgyz and Uzbek subpopulations. On average, *horizontal inequality* is the best predictor for displacement of Kyrgyz and for Uzbeks. Moreover, a strand of literature has looked at spatial segregation and how by deterring mutual trust and social capital, it might incite conflicts. As we do not have access to the addresses of households, we compute segregation at a higher granular level. Recent work uses national segregation indices in a cross-section of 91 countries to predict the intensity of conflict [12]. In line with this work, we measure the ethnic segregation of Uzbeks and Kyrgyz for each district $r$ with $S_r = \sum_j^J \sum_e^E t_{j,r} \frac{(n_{j,e,r} - \bar{n}_{e,r})^2}{\bar{n}_{e,r}}$. We define $\bar{n}_{e,r}$ as the population share of ethnicity $e$ on district-level $r$ and $n_{j,e,r}$ as the population share of ethnicity $e$ on community-level $j$. The segregation measure increases when ethnic group shares highly deviate across communities which pertain to the same district. Columns C and D of Table 5 report the

**Table 3. Forced displacement in communities.**

| | A | B | C | D | E | F |
|---|---|---|---|---|---|---|
| | b/se | b/se | b/se | b/se | b/se | b/se |
| Uzb.-Kyr. HI education 2% (0–1) | .54*** | .53*** | .54*** | .54*** | .61*** | |
| | (.19) | (.19) | (.19) | (.19) | (.22) | |
| Multi-group HI education (0–1) | | | | | | .49 |
| | | | | | | (.51) |
| Gini years of education (0–1) | | .2 | | .22 | .26 | .14 |
| | | (.2) | | (.36) | (.38) | (.32) |
| Gini education Uzbeks (0–1) | | | .24 | .23 | .2 | .26 |
| | | | (.29) | (.29) | (.29) | (.32) |
| Gini education Kyrgyz (0–1) | | | .08 | -.063 | -.098 | .0052 |
| | | | (.14) | (.29) | (.3) | (.26) |
| Ethnic polarization (0–1) | | | | | -.042 | -.021 |
| | | | | | (.034) | (.041) |
| Distance border Uzb. (10kms) | -.0011 | -.001 | -.001 | -.001 | -.00086 | -.0024** |
| | (.00074) | (.00076) | (.00081) | (.00081) | (.0008) | (.0011) |
| Total population (10000, 2009) | .0025 | .0026 | .0027 | .0028 | .0029 | .0031 |
| | (.0017) | (.0017) | (.0017) | (.0018) | (.0018) | (.0019) |
| Share born in community (%) | -.052 | -.042 | -.045 | -.04 | -.06 | -.05 |
| | (.061) | (.064) | (.068) | (.069) | (.077) | (.078) |
| Unemployment (%) | .11 | .092 | .1 | .075 | .15 | .045 |
| | (.16) | (.16) | (.16) | (.18) | (.2) | (.22) |
| Urban (yes/no) | .043 | .042 | .043 | .042 | .045 | .016 |
| | (.03) | (.029) | (.029) | (.029) | (.03) | (.028) |
| Major industry (yes/no) | .016 | .016 | .018 | .017 | .026 | .0094 |
| | (.019) | (.019) | (.019) | (.019) | (.02) | (.022) |
| Constant | -.04 | -.068 | -.078 | -.086 | -.064 | .0069 |
| | (.084) | (.096) | (.11) | (.11) | (.11) | (.12) |
| R2 | .34 | .34 | .35 | .35 | .36 | .24 |
| Observations | 120 | 120 | 120 | 120 | 120 | 120 |

*Note*: Sample based on 120 sampling communities in LiK study. Coefficients are average marginal effects reported from OLS regressions. Standard errors are robust. The dependent variable is the fraction of displaced population from the total population. Horizontal inequality defined in Eq (1) with Census 2009 data. Vertical inequality are gini coefficients as defined in section IV.D based on years of completed schooling from the LiK 2010 study. Ethnic polarization as defined in section IV.D based on Census data. Multi-group horizontal inequality as defined in section V.A.

\* $p < 0.1$,

\*\* $p < 0.05$,

\*\*\* $p < 0.01$.

Standard errors in parentheses.

coefficients from the sample of 51 districts and suggests that segregation does not lower the magnitude or significance of the HI index.

Outliers might be a concern suggested by the spatial concentration of violence and displacement in Fig 2. S3 Table in the S1 Appendix shows the robustness of associations in our base specification without Osh and Osh City. Moreover, the horizontal inequality index does not account for differences in group size. This can lead to extreme outcomes if for instance only one Uzbek lived in a community and her education would enter the index. We therefore assume an HI of 0 for communities where the Uzbek group size is below 2% in our base

**Table 4. Displacement in communities and violence.**

|  | A | B | C |
|---|---|---|---|
|  | Share Displaced | Share Displaced | Share Displaced |
|  | b/se | b/se | b/se |
| Uzb-Kyr HI education (0–1) | .45** | .55** | .44 |
|  | (.22) | (.25) | (.28) |
| Log dist. to violence | -.0093 | -.0067 | .011 |
|  | (.016) | (.018) | (.01) |
| Uzb-Kyr HI education (0–1) × Log dist. to violence |  |  | -.11 |
|  |  |  | (.085) |
| Gini years of education (0–1) |  | .19 | .24 |
|  |  | (.39) | (.37) |
| Gini education Uzbeks (0–1) |  | .18 | .17 |
|  |  | (.31) | (.31) |
| Gini education Kyrgyz (0–1) |  | -.064 | -.04 |
|  |  | (.29) | (.3) |
| Ethnic polarization (0–1) |  | -.042 | -.041 |
|  |  | (.034) | (.033) |
| Controls | Yes | Yes | Yes |
| R2 | .34 | .36 | .38 |
| N | 120 | 120 | 120 |

*Note*: Sample based on 120 sampling communities in LiK study after the conflict. Coefficients from linear model are reported. Horizontal inequality defined in Eq (1) with Census 2009 data. Vertical inequality are gini coefficients as defined in section IV.D based on years of completed schooling from the LiK 2010 study. Ethnic polarization as defined in section IV.D based on Census data. Multi-group horizontal inequality as defined in section V.A.

* $p < 0.1$,

** $p < 0.05$,

*** $p < 0.01$.

Standard errors in parentheses.

specifications. Nevertheless, in S4 Table in the S1 Appendix we document that the positive significant correlation is robust to using alternative group thresholds of 0%, 5% and 10% and using the group size-weighted multi-group measure of Community cross-section column F for only Kyrygz and Ubzeks. Moreover, S6–S8 Tables in the S1 Appendix in the appendix weigh the baseline results with the community share of ethnic Uzbeks and Kyrgyz of total Uzbek and Kyrgyz population and the share of total national population. Our qualitative interpretations remain by and large the same.

A source of concern about our estimates are omitted variables. Uncontrolled community characteristics related simultaneously to ethnic differences in education and to the decision to flee during the protest are problematic. For instance an imbalance in local political representation of ethnic groups could translate into unequal access to public goods such as schools, and community safety [44]. Language use across schools and regions might play a role in explaining education levels [45, 46]. Uzbeks could face less learning difficulties being taught exclusively in the Uzbek language, which could foster their cultural ties with Uzbekistan and thus the probability to flee across the border.

Another concern is that our displacement outcome might suffer from measurement error. Random measurement error would attenuate the coefficient down to zero. If displaced individuals would not return to their place of residence during the conflict because of ethnic inequalities, measurement error would act like an omitted variable bias. S5 Table in the S1

**Table 5. Forced displacement in communities and districts.**

| | A | B | C | D |
|---|---|---|---|---|
| | **Share Displaced Kyrgyz** | **Share Displaced Uzbek** | **Share Displaced** | **Share Displaced** |
| | b/se | b/se | b/se | b/se |
| Uzb-Kyr HI education (0–1) | .359** | .567** | .605** | .685* |
| | (.164) | (.218) | (.294) | (.358) |
| Gini years of education (0–1) | .418 | .493 | -.0955 | -.0875 |
| | (.297) | (.53) | (.221) | (.206) |
| Gini education Uzbeks (0–1) | .0714 | .653 | | |
| | (.299) | (.408) | | |
| Gini education Kyrgyz (0–1) | -.282 | -.199 | | |
| | (.262) | (.376) | | |
| Ethnic polarization (0–1) | -.0128 | -.0205 | -.0201 | -.0211 |
| | (.0264) | (.0276) | (.0235) | (.0224) |
| Segregation (index, 0–1) | | | | -.0654 |
| | | | | (.0559) |
| Controls | Yes | Yes | Yes | Yes |
| Granularity | Community | Community | District | District |
| R2 | .313 | .339 | .445 | .48 |
| Observations | 120 | 120 | 51 | 51 |

*Note*: Columns A-B are estimated with the full sample. Columns C and D use the same sample aggregated at the rayon level. All columns report coefficients from linear regressions. Horizontal inequality defined in Eq (1) with Census 2009 data. Vertical inequality are gini coefficients as defined in section IV.D based on years of completed schooling from the LiK 2010 study. Ethnic polarization as defined in section IV.D based on Census data. Segregation is measured as defined in V.A.

* $p < 0.1$,

** $p < 0.05$,

*** $p < 0.01$.

Standard errors in parentheses.

Appendix reports results with the fraction of displaced among those who indicate to have never lived somewhere else. This subsample of the population is certainly different in many aspects, but estimated coefficients are very similar in magnitude and significance. This could be interpreted that bias from returning to live somewhere else in Kyrgyzstan after forced displacement must be limited. Moreover, there might be more non-returnees to Kyrgyzstan in conflict-prone communities. This could create confounding correlated with ethnic inequalities as well.

To quantify the sensitivity of these results with respect to omitted variables we report a battery of statistics for the full baseline specification in Table 3 column E [47]. Table 6 shows the robustness value $RV_{q = 1}$ which indicates that the level of association for confounders with HIs and displacement needed in order to overturn the coefficient to 0 is 40.4%. Additionally, $R^2_{Y \ D|X}$ measures the level of association of confounders with HIs needed to bring the coefficient of HIs to 0 is 21.5%, under the bounding assumption that confounders explained all of the residual variance of the regression. To benchmark the magnitude of these values, we report the partial $R^2$ of distance to the Uzbek border with the outcome ($R^2_{Y \sim Z|D,X} = 2\%$) and HIs ($R^2_{D \sim Z|X} = 24.3\%$) below the table. As the robustness value of 40.4% is greater than both values, we can summarize that a confounder with the strength of association of the distance to the Uzbek border would not overturn the main results.

**Table 6. Sensitivity: Community cross-section.**

| Outcome: *Forced Displacement* | | | | | | |
|---|---|---|---|---|---|---|
| Explanatory Variable: | Est. | S.E. | t-value | $R^2_{Y \sim D|\mathbf{X}}$ | $RV_{q=1}$ | $RV_{q=1, \alpha=0.05}$ |
| *Uzb-Kyr HI education(0–1)* | 0.608 | 0.112 | 5.442 | 21.5% | 40.4% | 28.1% |
| df = 108 | *Bound (1x Dist. to Uz. border):* $R^2_{Y \sim Z|\mathbf{X}, D} = 2\%$, $R^2_{D \sim Z|\mathbf{X}} = 24.3\%$ | | | | | |

*Note*: Sensitivity statistics computed from the regression in Table 3 column E [47]. Column 4 indicates that in the case where unobserved confounders explain all of the residual variance of the outcome, confounders would have to explain the percentage in column 4 of the residual variance of treatment to bring the coefficient of HI to 0. Column 5 contains the robustness value to bring the coefficient of HIs down to 0. Column 6 reports the robustness value that the p-value of the same coefficient increases beyond 5%. Bounds on confounding are reported below the table with respect to the distance to the Uzbek border.

In summary, *horizontal inequality* emerges as a robust predictor of displacement across communities in the Kyrgyz context. These results do not support the view, where conflict arises from within-ethnic-group inequality splitting an ethnic group into rich providers of financial means and poor suppliers of conflict labor [48]. We would rather argue along the lines of past work that on the one hand, the collective experience of deprivation from access to basic and higher education might spur protest mobilization [26]. On the other hand, privileged access to education provided on the basis of ethnicity might strengthen it's importance for identity. Our results are surprising in the sense that Kyrgyzstan has a relatively upward mobile education system in international comparison [49]. Education is publicly financed and and there exists for example a nationwide policy of free school meals [50]. Notwithstanding Kyrgyzstan's historically egalitarian education system, one might argue that competition through more accessible education might make discrimination on the basis of ethnicity even more attractive to a local elite to maintain their status. Anecdotal evidence describes how Uzbek was scrapped as an official language to take state exams in a process described as the "Kyrgyzification" of the teaching system [51].

## 5.2 Individual-level regressions

Exploiting geographic variation, the previous section suggests that more people flee from areas with unequal educational outcomes between ethnic groups. This section seeks to find out who these people are comparing their education and ethnicity with others in the place they live. We exploit variation within districts by regressing the decision to flee on the individual's position in the local joint education and ethnicity distribution. We assume linear association of education ranks on displacement through the usage of simple linear probability models. A question we would like to answer is: is an ethnic Uzbek person more likely to leave when her educational attainment is above the Kyrgyz' district average? We estimate the following regression:

$$
\begin{aligned}
P(D_{i,j} = 1) \quad &= \alpha + \beta_1 uz_i + \beta_2 ky + \gamma_1 \mathbf{1}(\mathrm{edu}_{i,j} > t_j^k) + \gamma_2 \mathbf{1}(\mathrm{edu}_{i,j} > t_j^u) \\
&\quad + \delta_1 \, uz_i \, \mathbf{1}(\mathrm{edu}_{i,j} > t_j^k) + \delta_2 \, ky_i \, \mathbf{1}(\mathrm{edu}_{i,j} > t_j^u) \qquad (3) \\
&\quad + \pi edu_i + \beta' \mathbf{x}_{i,j} + d_j + \epsilon_{i,j}
\end{aligned}
$$

The dummy variable $\mathbf{1}(\mathrm{edu}_{i,j} > t_j^e)$ in Eq 3 indicates whether individual $i$ with education level $edu_i$ is above a threshold $t_j^e$ on the distribution of education of the other ethnic group $e$ (Kyrgyz or Uzbek) in area $j$. We wish to estimate whether e.g. Uzbeks who are either more or less educated than most Kyrgyz in their district might have different probabilities of fleeing

**Table 7. Individual displacement and percentiles.**

|  | A | B | C | D | E |
|---|---|---|---|---|---|
|  | b/se | b/se | b/se | b/se | b/se |
| Education(completed years) | .00023 | -.00046 | -.0008 | .000034 | .00002 |
|  | (.00063) | (.0008) | (.00078) | (.00093) | (.00071) |
| Uzbek (No = 0, Yes = 1) | .036*** | .05*** | .075*** | .1*** | .15*** |
|  | (.011) | (.018) | (.028) | (.031) | (.045) |
| Krygyz (No = 0, Yes = 1) | .0055 | .0081 | -.0048 | .011 | .024** |
|  | (.0038) | (.0058) | (.0069) | (.0085) | (.011) |
| $1(\text{edu} > p^{uz})$ |  | .012 | -.02 | -.0016 | .016 |
|  |  | (.018) | (.013) | (.015) | (.019) |
| $1(\text{edu} > p^{ky})$ |  | -.0081 | -.0035 | .01* | .011 |
|  |  | (.0061) | (.0044) | (.0063) | (.0072) |
| HI: Uzbek * $1(\text{edu} > p^{ky})$ |  | -.021 | -.039 | -.075** | -.12** |
|  |  | (.025) | (.031) | (.035) | (.048) |
| HI: Kyrgyz * $1(\text{edu} > p^{uz})$ |  | -.0089 | .023* | -.013 | -.028* |
|  |  | (.016) | (.012) | (.013) | (.016) |
| Controls | Yes | Yes | Yes | Yes | Yes |
| Threshold statistic | - | 20th | 40th | 60th | 80th |
| Threshold geography $j$ | - | District | District | District | District |
| Regional control | District | District | District | District | District |
| N | 6993 | 6993 | 6993 | 6993 | 6993 |

*Note*: Linear probability models with binary outcome equal to 1 if individual was displaced. The sample is based on all individual respondents of the LiK study older than 17 years. Positional inequality dummy $1(\text{edu} > t)$ are as defined in Eq 2. As thresholds we use quintile values as specified in row Threshold stat. Controls are distance to Uzbek border, age bins, gender, nr of children, married, parental education background, self-employment, born in community and landowner. All specifications include dummy variables for each district.

* p < 0.1,

** p < 0.05,

*** p < 0.01.

Standard errors in parentheses.

controlling for the general influence of education. In practice we use percentiles of the local ethnicity-specific education distribution (quintiles). This positional dummy variable is used in interaction with ethnicity dummies to capture relations with ethnic group identity. By including district dummmy variables $d_j$, we control for the level of thresholds and exploit variation within district $j$. We control for confounding factors with a vector of controls $\mathbf{x}_{i,j}$. Standard errors are robust to heteroskedasticity [42].

We interpret the $\delta$ coefficients as average associations from between-ethnicity differences in education on the probability to flee. We define *horizontal inequality* for Kyrgyz as the interaction $ky_i * 1(\text{edu}_{i,j} > t_j^u)$ and for the case of Uzbeks as $uz_i * 1(\text{edu}_{i,j} > t_j^k)$. We think of these interactions as an indicator of "educational advantage" towards some fraction of the other ethnic group. For example when an ethnic Uzbek is more educated than the median Kyrgyz in her district, she would therefore be "advantaged" vis à vis half of the Kyrgyz living in her district. Note that the omitted category are all other ethnicities. In mono-ethnic districts we set $1(\text{edu}_{i,j} > t_j^e)$ to 0.

Table 7 summarizes our regression results. Specification A reports the baseline estimates of forced displacement on education and ethnicity dummies. In columns B through E, we

introduce the dummies indicating whether an individual was above a specific quintile of the distribution of education of the other ethnicity.

Column A in Table 7 shows that self-identifying as ethnic Kyrgyz was not more likely to be associated with displacement during the conflict compared to Uzbeks or the base category of all other minorities (e.g. Russians, Uigurs). We control for unobservables at the district level and for education, age dummies, female, number of children, marital status, education of father, self employment, land owner, born in community and distance to border. However, Uzbek ethnicity is associated with a probability of displacement, $\beta_1 = 3.6pp$ higher than other minorities in the base category. Introducing horizontal inequality, columns B to E reveal how horizontal disadvantage in education towards the other ethnic groups manifests itself in an associated increase in the likelihood to flee. On the other hand, horizontal advantage of Uzbeks and Krygyz vis à vis the other ethnic group is associated with a lower probability to flee. Uzbeks better educated than the bottom 80% of Kyrgyz in the district are $\delta_1 = -12pp$ less likely to be displaced than Uzbeks with education levels comparable to the bottom 80% of Kyrgyz. The same direction of correlation holds true for ethnic Kyrgyz who are $\delta_2 = -2.8pp$ less likely to be displaced above than below the 80th percentile of Uzbek education in their district. Compared to individuals of other ethnicites than Uzbek and Kyrgyz, Uzbeks ranked in the bottom 80% of the local Kyrgyz distribution have a $\beta_1 = 15pp$ higher probability of displacement. Analogously, low educated Kyrgyz ranked in the bottom 80% of the Uzbek distribution have a $\beta_2 = 2.8pp$ higher probability of displacement than other ethnicities. This means that both highly ranked Uzbeks and Kzrgyz relative to the other ethnic group were significantly less likely to flee, even though the association is much stronger for Uzbeks. The association of educational advantage does not hold for other ethnic groups as evidenced by the insignificant coefficients $\gamma_1$ and $\gamma_2$. We emphasize that we additionally control for education, which implies that not simply high education levels are associated with lower probabilities to flee.

Table 8 includes interactions in the specifications reported in Table 7 with the distance to casualties. A 100% percent decrease in the distance to violence is associated with a 2.3–2.5ppt increase in the base probability of displacement throughout all specifications. Horizontal advantage alone and in interaction with distance to violence are both significant for Uzbeks. Columns D and E show that an Uzbek more educated than most Krygyz in the district he or she lives in is more likely to stay than an Uzbek with lower education. The positive coefficient of interaction implies that this effect is more pronounced the closer the highly educated Uzbek lives to the center of violence. According to our theory, violence leads to displacement through mechanism M3. The fact that horizontally disadvantaged individuals are significantly more likely to be displaced if violence occurs in their proximity might be explained through two potential mechanism. Firstly, educational advantage might entail superior information about the conflict. The informational advantage might help to navigate public spaces or find protection by law enforcement. Secondly, horizontal advantaged are more likely to speak Uzbek or Kyrgyz as a second language and better networked to communities across ethnic barriers. They might have been in a better position to rely on interethnic networks to find shelter during the conflict [52]. In theory, a direct link between inequalities and displacement exists through mechanism M2. The fact that horizontal advantage remains significant in the interacted model with violence could be interpreted as suggestive evidence for this link. Why could inequalities still lead to displacement even though casualties did not occur in their districts? Firstly, simple answer might suggest that there is unobserved violence not necessarily correlated with casualties such as non-lethal physical or psychological violence, discrimination or sexual abuse. Misinformation about the alleged rape of a young Kyrgyz student by Uzbek men mobilized many Kyrgyz men to engage in violence against Uzbeks [18]. Due to the stigmatization of victims, international observers judge the assessment of sexual violence generally difficult in

**Table 8. Individual displacement and percentile.**

| | A | B | C | D | E |
|---|---|---|---|---|---|
| | b/se | b/se | b/se | b/se | b/se |
| Education(completed years) | -.000092 | -.0009 | -.001 | 9.5e-06 | -.00016 |
| | (.00062) | (.00079) | (.00079) | (.00092) | (.0007) |
| Uzbek (No = 0,Yes = 1) = 1 | .11*** | .17*** | .2*** | .3*** | .48*** |
| | (.042) | (.065) | (.075) | (.1) | (.17) |
| Krygyz (No = 0,Yes = 1) = 1 | .13*** | .14*** | .12*** | .14*** | .15*** |
| | (.042) | (.043) | (.044) | (.045) | (.046) |
| $1(edu > p^{uz}) = 1$ | | -.0041 | -.024* | -.0018 | .021 |
| | | (.018) | (.014) | (.015) | (.023) |
| $1(edu > p^{ky}) = 1$ | | -.0067 | -.0012 | .0093 | .0076 |
| | | (.0061) | (.0044) | (.0058) | (.0065) |
| HI: Uzb. * $1(edu > p^{ky}) = 1$ | | -.065 | -.08 | -.19** | -.38** |
| | | (.057) | (.068) | (.095) | (.17) |
| HI: Kyr. * $1(edu > p^{uz}) = 1$ | | -.023 | .038 | -.015 | -.04 |
| | | (.026) | (.033) | (.034) | (.043) |
| Lg. dist. viol. | -.024** | -.024** | -.024** | -.023** | -.025** |
| | (.01) | (.01) | (.01) | (.011) | (.011) |
| Uzbek (No = 0,Yes = 1) = 1 × Lg. dist. viol. | -.0079 | -.024* | -.023 | -.043** | -.081** |
| | (.0084) | (.014) | (.016) | (.019) | (.036) |
| Krygyz (No = 0,Yes = 1) = 1 × Lg. dist. viol. | -.025*** | -.027*** | -.024*** | -.026*** | -.027*** |
| | (.0077) | (.0079) | (.0081) | (.0082) | (.0084) |
| HI: Kyr. * $1(edu > p^{uz}) = 1$ × Lg. dist. viol. | | .0073* | -.0034 | .0028 | .0057 |
| | | (.0044) | (.0063) | (.006) | (.0073) |
| HI: Uzb. * $1(edu > p^{ky}) = 1$ × Lg. dist. viol. | | .021 | .016 | .035* | .073** |
| | | (.014) | (.016) | (.019) | (.036) |
| Controls | Yes | Yes | Yes | Yes | Yes |
| Threshold statistic | - | 20th | 40th | 60th | 80th |
| Threshold geography *j* | - | District | District | District | District |
| Regional control | District | District | District | District | District |
| N | 6993 | 6993 | 6993 | 6993 | 6993 |

*Note*: Linear probability models with binary outcome equal to 1 if individual was displaced. The sample is based on all individual respondents of the LiK study older than 17 years. Positional inequality dummy $1(edu > t)$ are as defined in Eq 2. As thresholds we use quintile values as specified in row Threshold stat. Controls are distance to Uzbek border, age bins, gender, nr of children, married, parental education background, self-employment, born in community and landowner. All specifications include dummy variables for each district.

* p < 0.1,

** p < 0.05,

*** p < 0.01.

Standard errors in parentheses.

Kyrgyzstan [18]. However, journalists documented anecdotal evidence of rape by men against women of the other ethnicity, in particular against Uzbek women [53]. One could rationalize the decision to flee taken by many horizontally disadvantaged women as a measure to protect themselves and household dependents from the escalating violence between men of both ethnic groups. Secondly, a subjective assessment of risk in ethnic conflict might not necessarily be based on where fighting occurs, but rather how disadvantaged individuals feel relative to members of the other ethnic group. Individuals might not necessarily observe more violence than

the statistician, an individual's position objective position in the social hierarchy might explain a subjective assessment of their risk to be targeted by ethnic armed groups.

We perform a battery of robustness checks. A concern for the individual regressions is that survey respondents might have lived somewhere else within Kyrgyzstan prior to the conflict than the place where they were surveyed. S9 Table in the S1 Appendix reproduces Individual-level regressions for a subsample of individuals who have always lived in the community they were interviewed in. The results hold, except that the coefficients for ethnic Kyrgyz are not significant anymore at 5%. S10 Table in the S1 Appendix breaks the outcome down in cross-border and internal replacement. Cross-border displacement is only associated with Uzbeks.

## 6 Conclusion

This paper studies the relationship between forced displacement and ethnic inequalities during a violent conflict. The case of Kyrgyzstan provides a rare opportunity to exploit a nationally representative survey collected several months after a violent episode, which displaced mainly members of the ethnic minority Uzbeks, but also members of its ethnic majority the Kyrgyz.

Comparing districts we find that districts with higher inequalities in education between both ethnic groups had higher displacement rates. Comparing individuals, we observe that those with an educational advantage at the local level towards the other group had a lower average probability to be displaced. Our findings suggest that displacement increases with the proximity to deaths, but adjusting for distance to violence, inequalities still predict high displacement. Our findings therefore demonstrate that regions with large disparities between ethnic groups might bear the highest burden from the conflict. Concerning peace-making policy, our findings suggest that inequalities need to be addressed at the local level.

We hope that future work can circumvent some of the identification assumptions in this paper imposed by data availability. Following individuals through the conflict and potential displacement over time with longitudinal data would open a new set of possibilities to study the decision to flee from conflict.

## Supporting information

**S1 Appendix.**
(PDF)

## Acknowledgments

We thank Gani Aldashev, Damir Esenaliev, Neil Ferguson, Roman Mogilevskii, Susan Steiner and participants at the Life in Kyrgyzstan Conference in Bishkek and the 12th HiCN Workshop at the FAO in Rome.

## Author Contributions

**Conceptualization:** Moritz Hennicke, Tilman Brück.

**Data curation:** Moritz Hennicke, Tilman Brück.

**Formal analysis:** Moritz Hennicke, Tilman Brück.

**Funding acquisition:** Moritz Hennicke, Tilman Brück.

**Investigation:** Moritz Hennicke, Tilman Brück.

**Methodology:** Moritz Hennicke, Tilman Brück.

**Project administration:** Moritz Hennicke, Tilman Brück.

**Resources:** Moritz Hennicke, Tilman Brück.

**Software:** Moritz Hennicke, Tilman Brück.

**Supervision:** Moritz Hennicke, Tilman Brück.

**Validation:** Moritz Hennicke, Tilman Brück.

**Visualization:** Moritz Hennicke, Tilman Brück.

**Writing – original draft:** Moritz Hennicke, Tilman Brück.

**Writing – review & editing:** Moritz Hennicke, Tilman Brück.

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
