## [Decision Letter · Decision Letter 0]

9 Sep 2021

PONE-D-21-18965Ethnic Inequality and Forced DisplacementPLOS ONE

Dear Dr. Hennicke,

Thank you for submitting your manuscript to PLOS ONE. After careful consideration, we feel that it has merit but does not fully meet PLOS ONE’s publication criteria as it currently stands. Therefore, we invite you to submit a revised version of the manuscript that addresses the points raised during the review process.

The reviewers have opposing views. Reviewer #1 requests revisions, while Reviewer #2 is not so positive about your paper. I read the paper myself; I believe that the topic is very interesting and the paper has potential. Therefore, I would like to give you a chance to address the reviewers' comments. That said, I want to emphasize that eventual publication of the revised manuscript is not guaranteed. If you choose to revise the paper, please try to address the reviewers' comments as fully as you can.

We look forward to receiving your revised manuscript.

Kind regards,

Semih Tumen, PhD

Academic Editor

PLOS ONE

Journal Requirements:

5. We note that you have referenced (ie. Bewick et al. [5]) which has currently not yet been accepted for publication. Please remove this from your References and amend this to state in the body of your manuscript: (ie “Bewick et al. [Unpublished]”) as detailed online in our guide for authors

6. We note that Figure 1 in your submission contain [map/satellite] images which may be copyrighted. All PLOS content is published under the Creative Commons Attribution License (CC BY 4.0), which means that the manuscript, images, and Supporting Information files will be freely available online, and any third party is permitted to access, download, copy, distribute, and use these materials in any way, even commercially, with proper attribution. For these reasons, we cannot publish previously copyrighted maps or satellite images created using proprietary data, such as Google software (Google Maps, Street View, and Earth). For more information, see our copyright guidelines: http://journals.plos.org/plosone/s/licenses-and-copyright.

Reviewers' comments:

Reviewer's Responses to Questions

**Comments to the Author**

1. Is the manuscript technically sound, and do the data support the conclusions?

Reviewer #1: Partly

Reviewer #2: Partly

2. Has the statistical analysis been performed appropriately and rigorously? 

Reviewer #1: Yes

Reviewer #2: I Don't Know

3. Have the authors made all data underlying the findings in their manuscript fully available?

Reviewer #1: Yes

Reviewer #2: No

4. Is the manuscript presented in an intelligible fashion and written in standard English?

Reviewer #1: Yes

Reviewer #2: Yes

5. Review Comments to the Author

Reviewer #1: Please see uploaded document, which includes the entire review prepared for the paper, "Ethic inequality and forced displacement". The paper uses the Life in Kyrgyzstan study and census data to examine how inequality between and within ethnic groups relates to displacement.

Reviewer #2: Ethnic Inequality and Forced Displacement

This paper studies an important question, namely, what is the role of ethnic inequality in forced displacement. This paper employs interesting data and explores an interesting and relevant context: conflict in Kyrgyzstan and its consequences on ethnic Uzbeks. I believe it is increasingly important to have evidence from various contexts and learned a lot from reading this paper. The paper presents evidence that suggests that there is an important correlation between ethnic inequality in education and forced displacement. As it is, the paper lacks a proper theoretical discussion that can inform the empirical exercise. Although the authors note there are identification issues, they are extremely under-discussed. These results could be interesting even if only observational, but we need a theory for many reasons a) to understand why we expect or not to observe the correlation the authors find, b) the mechanisms that drive the correlation, to underscore the policy relevance of the paper, and c) to at least think about the many potential confounders that may be behind the correlation. Below I provide some comments and suggestions that hopefully will make this paper stronger.

Theory

The paper has no theory. From the beginning, the authors talk about the importance of ethnicity in conflict but fail to provide a clear theoretical link between inequality in education observed across ethnic lines and forced displacement. This makes the paper extremely confusing and leaves the reader wondering why we should observe any correlation at all. This is further complicated by the fact that the authors list an extensive list of papers about ethnicity and conflict, but do not discuss at all how they relate to their work, or what they contribute to the literature. I suggest that the authors add a theory section where they lay out what is the relationship between the variables they discuss. It seems to me (but I may be confused) that the authors have in mind a causal process that goes from ethnicity to conflict, and has displacement as an outcome, however, the relationship is mediated by education. Laying out these links could also be helpful to note that there is an obvious confounder that affects both inequality in education and the probability of being displaced, which is barely mentioned: income. Several scholars have written extensively about this relationship, and about ethnic minorities and displacement in Colombia, and none are mentioned. Some starting points could be the work of Ana Maria Ibañez or Abbey Steele. More on data later, but the authors should control for income.

Additionally, there are a lot of things we don’t know about the context that could be useful to understand how income affects inequality and vice-versa (v.gr. do Uzbeks have schools where they speak their own language? Is it hard to integrate into the labor market if you don’t speak the local language?). Moreover, a theory could be helpful to understand the mechanisms that are underpinning the relationship, because the reader is unable to grasp what is going on: are the less educated being targeted because there are poorly educated and forced to leave? Are the less educated being targeted because they are poor (and being poorly educated is correlated with income)? Are they not being targeted, and leaving out of their own volition to avoid risks? Are they living because they are afraid that they don’t have access to the judicial system? Because we don’t understand what is going on, the empirical exercise is left to stand alone, and the reader is obliged to make sense of the results. Although in the empirical section the authors affirm, they are not making causal claims, the paper is written as if they were until that section. A theoretical explanation of why income is not a confounder should alleviate that concern partly. I recommend that a proper (and by this, I don’t mean necessarily formal) theoretical framework is laid out after the context is presented to explain the relationship of the main variables of interest (is education a moderator??), and the main mechanisms.

Data

It is hard to understand how the data being used can capture what the author is trying to measure. In particular, the data is post-conflict: don’t we expect the makeup of the population left behind to be very different from the population pre-conflict? In particular: suppose the wealthy left, aren’t we not overestimating the responses from the poor (and less educated) left behind? I believe the authors should discuss how this is not biasing the results, although I can’t think of any process that would not generate sample selection bias. If the authors believe the conflict-affected families randomly, and that a random subset of the Uzbeks was left behind such that we can say something that generalizes, they should discuss it.

The authors should discuss in what ways the measure HI is not problematic, given that the measure does not account for different group sizes. Maybe they could weigh the districts by group size? Moreover, the authors should explain why then they seem to test inequality among multiple groups with a new measure that does give group size importance. Why not use the same original measure and control by each group’s measure?

A special mention goes to the fact that none of the figures or tables are references, and that the famous Figure 1 is either absent or the map presented at the end, which does not show what the authors claim it shows, as there are no labels and it is very hard to make sense of.

Empirical strategy

The explanation for using a glm model is not clear: there are several corrections to estimate standard errors when errors are not normally distributed using OLS. The authors should try to implement these corrections and show an OLS estimation as robustness. The model used in the individual displacement section seems more convincing.

The authors should interpret the coefficients in the probability of staying for educated groups of both predominant ethnicities since this brings even more doubts: the fact that both educated Uzbeks and Kyrgyz are more likely to stay makes the reader wonder if what you are showing is not a correlation between income and displacement, which is very likely to be the case.

Misc

The paper should be edit for clarity.

6. PLOS authors have the option to publish the peer review history of their article (what does this mean?). If published, this will include your full peer review and any attached files.

Reviewer #1: No

Reviewer #2: No

---

## [Author Response · Author response to Decision Letter 0]

16 Nov 2021

Please see the attached letter for our point by point reply to the Reviewers. Thank you.

---

## [Decision Letter · Decision Letter 1]

24 Jan 2022

PONE-D-21-18965R1Ethnic Inequality and Forced DisplacementPLOS ONE

Dear Dr. Hennicke,

Thank you for submitting your manuscript to PLOS ONE. After careful consideration, we feel that it has merit but does not fully meet PLOS ONE’s publication criteria as it currently stands. Therefore, we invite you to submit a revised version of the manuscript that addresses the points raised during the review process.

Although the decision is still "Major Revision," I am happy to see that you have at least partly convinced the negative referee; and, s/he also asks revisions in this round. S/he particularly asks further and more convincing elaboration on the theoretical underpinnings of the estimates/correlations you present. I agree with his/her views. This means that the manuscript has now started to converge toward a favorable outcome, but still there is no guarantee for final acceptance at this point.

We look forward to receiving your revised manuscript.

Kind regards,

Semih Tumen, PhD

Academic Editor

PLOS ONE

Reviewers' comments:

Reviewer's Responses to Questions

**Comments to the Author**

1. If the authors have adequately addressed your comments raised in a previous round of review and you feel that this manuscript is now acceptable for publication, you may indicate that here to bypass the “Comments to the Author” section, enter your conflict of interest statement in the “Confidential to Editor” section, and submit your "Accept" recommendation.

Reviewer #1: All comments have been addressed

Reviewer #2: (No Response)

2. Is the manuscript technically sound, and do the data support the conclusions?

Reviewer #1: Yes

Reviewer #2: Partly

3. Has the statistical analysis been performed appropriately and rigorously? 

Reviewer #1: Yes

Reviewer #2: N/A

4. Have the authors made all data underlying the findings in their manuscript fully available?

Reviewer #1: Yes

Reviewer #2: Yes

5. Is the manuscript presented in an intelligible fashion and written in standard English?

Reviewer #1: Yes

Reviewer #2: Yes

6. Review Comments to the Author

Reviewer #1: It is very clear from the updated manuscript and the authors’ letter to the reviewers that a great deal of time, thought, and care went into the revisions. I thank the authors for these considerable efforts, which have certainly made the manuscript stronger. One major improvement is that the theoretical channels linking inequality, violence, and displacement are laid out much more clearly for the reader. I also very much appreciated how the authors used the theoretical framework to invoke relevant past literature. I also greatly appreciate the authors’ careful empirical treatment of the challenges associated with the inter-relationships between the inequality, conflict, and displacement variables.

I want to thank the authors for the tremendous time and care that they put into these revisions and for producing this meaningful research. To be honest, most of my comments relate to cleaning the manuscript up a little more.

MAJOR COMMENTS

The literature review and theory sections have improved tremendously. I have one suggestion for enhancing this section further. One of the determinants of conflict behavior often discussed in economics is the opportunity cost of violence (a concept with origins in Gary Becker’s work on crime). I wonder if, given the use of education to operationalize inequality, the opportunity cost of violence may also play a role in understanding the relationship between horizontal inequality and conflict. For example, could the group that has received fewer educational opportunities have a lower opportunity cost of violence? Drawing that literature into the paper’s literature review could further enhance the manuscript.

Overall, the code used to render the manuscript (in Latex, I am guessing?) will need to go through a careful edit in order to ensure that sections are in their appropriate position and tables are not repeated. I was able to review two separate versions, a new clean manuscript and a manuscript showing tracked changes. I focused on the latter, as it helped me understand where changes had taken place. But in both versions, there were some organizational issues in terms of where tables, figures, and sometimes even sections appeared, as well as some repetition of Tables (ex: Table 3 and 4 in the track-edits version are the same?). It’s difficult for me to identify specific fixes since the problems may be slightly different across the two versions I am reading. Please address this.

I consider this a non-essential suggestion (please take it or leave it), but as a reader I was curious about what the authors saw as the future direction of this work on inequality and displacement. What additional questions or ideas did your analysis of Kyrgyzstan spark? Please don’t feel obligated to fabricate this part (sometimes the “next steps” section of a paper feels like total fluff), but if the authors have some ideas for future directions to add to the Conclusion, I would be super interested in reading this.

MINOR COMMENTS (mostly typos and formatting)

P. 3 “Northern capital Bishkek” → “northern capital Bishkek”

Fig 1 (clean version) formatting issue: with a black background, black text that exceeds the white shape size is lost and arrows are not visible. Please fix.

Line 248, 256, perhaps other lines - starts with a period, please remove.

Line 283 spelling/style suggestion: “Our work relates to past work, which systematically measure differentials” → “Our work relates to past work that systematically measures differentials”

Figure 3: Please upper case the ethnic group names on the X axis

Figure 3: Please add percentages above the bars to help with interpretation.

Figure 3: The figure is a bit hard to read since it’s unclear how the distributions add up to 100%. I recommend restructuring the graph so that each ethnicity-displacement group’s bars are stacked and add up to 100%. For example, there would be one stacked bar for non-displaced Kyrgyz that shows the distribution of educational attainment for this group. The figure can then report in the Notes what percent of each group was displaced.

Line 371 - typo: “Osh city” → “Osh City”

Line 393 - typo: “ist” → “is”

Fig. 4 (and elsewhere when relevant) - I think it would look a little better if the shorthand names for ethnicities (uzb) were upper-cased with a period at the end to show its an abbreviation (Uzb.).

Reviewer #2: I appreciate the effort in answering to the concerns previously stated. The paper has now a theory section that attempts to clarify what are the mechanisms behind the correlations they present. However, I believe there is still potential to present a clearer theoretical framework.

The finding that people with lower education levels flee more often is not novel, as I stated in previous report, several scholars have studied the determinants of displacement and found income, education, and vulnerability to be the main drivers. The authors should exploit this opportunity to explain what is going on behind that correlation, which I believe they don’t sufficiently do yet. To put it briefly, the authors say: “We find that educational advantage towards the other ethnic group in the area was associated with a lower probability to flee.” Why is this the case? More educated people are more powerful, have higher incomes, are part of the bureaucracy? Have better-defined property rights? Wouldn’t these people be precisely the ones that you want to go away? If not, why not?

The lack of clarity on this points, which would give the paper more bandwidth in terms of audience and policy-implications, stems from my view from several issues that are not addressed.

1. Forced displacement is often strategic, not just the result of the cross fire (plenty of qualitative and quantitative work has been written, including Abbey Steele, Antonella Bandiera, Ana María Ibañez, some which you cite). What is going on here? People have a choice? Why do the the less educated don’t have a choice? Someone is forcing them to leave? They don’t own property? The place they inhabit can be appropriated?

2. The current theory is confusing. The authors state there is direct link between ethnic inequality and displacement, that is not mediated by conflict, and they say “We conjecture that, besides violence, a direct connection (ethnic inequalities → displacement) exists and operates through mechanism M2. On the one hand, members of the discriminated group might be able to access better educational opportunities after fleeing. On the other hand, members of the better educated group might be more ready to flee expecting better opportunities for them at arrival.” It is hard to understand why they refer to this as displacement, when it sounds it is just voluntary migration. If you have better prospects and mobility is possible, you migrate. However, if the point is that conflict deteriorates economic opportunities and then you are forced to leave, then the relationship is necessarily mediated by conflict.

Lastly, it is hard to understand what is the role of conflict, partly because of the previous points, but also because you don't look at any heterogenous effects by intensity of conflict, or attempt to control using for example a pre-treatment variable that predicts intensity.

I recommend that the authos re-assess the theoretical analysis: it is still not clear why we should expect more educated people to stay (there are many possible reasons, but the ones you briefly mention are not related to conflict or inequality, rather to the existence of potential economic opportunities elsewhere). Although you have made great progress, without a clear theory that links ethnic inequality in education to displacement (not migration!), we are left with some correlations that are useful but do not allow us to derive any significant policy implications.

7. PLOS authors have the option to publish the peer review history of their article (what does this mean?). If published, this will include your full peer review and any attached files.

Reviewer #1: No

Reviewer #2: No

---

## [Author Response · Author response to Decision Letter 1]

18 Mar 2022

Please see attached pdf. Thank you.

---

## [Editor Report · Decision Letter 2]

22 Mar 2022

Ethnic Inequality and Forced Displacement

PONE-D-21-18965R2

Dear Dr. Hennicke,

We’re pleased to inform you that your manuscript has been judged scientifically suitable for publication and will be formally accepted for publication once it meets all outstanding technical requirements.

Kind regards,

Semih Tumen, PhD

Academic Editor

PLOS ONE
---

## [Editor Report · Acceptance letter]

31 Mar 2022

PONE-D-21-18965R2 

Ethnic inequality and forced displacement 

Dear Dr. Hennicke:

I'm pleased to inform you that your manuscript has been deemed suitable for publication in PLOS ONE. Congratulations! Your manuscript is now with our production department. 

Kind regards, 

on behalf of

Professor Semih Tumen 

Academic Editor

PLOS ONE